# Strategies for Traceability to Prevent Unauthorised GMOs (Including NGTs) in the EU: State of the Art and Possible Alternative Approaches

**DOI:** 10.3390/foods13030369

**Published:** 2024-01-23

**Authors:** Jenny Teufel, Viviana López Hernández, Anita Greiter, Nele Kampffmeyer, Inga Hilbert, Michael Eckerstorfer, Frank Narendja, Andreas Heissenberger, Samson Simon

**Affiliations:** 1Öko-Institut e.V., Merzhauser Strasse 173, 79100 Freiburg, Germany; v.lopez@oeko.de (V.L.H.); n.kampffmeyer@oeko.de (N.K.); i.hilbert@oeko.de (I.H.); 2Environment Agency Austria, Spittelauer Lände 5, 1090 Vienna, Austria; anita.greiter@umweltbundesamt.at (A.G.); michael.eckerstorfer@umweltbundesamt.at (M.E.); frank.narendja@umweltbundesamt.at (F.N.); andreas.heissenberger@umweltbundesamt.at (A.H.); 3Federal Agency for Nature Conservation, Konstantinstraße 110, 53179 Bonn, Germany; samson.simon@bfn.de

**Keywords:** genetically modified organisms (GMOs), new genomic techniques (NGTs), traceability systems, non-GMO, due diligence, sustainable development, transparency, European Union (EU), regulatory frameworks, compliance, risk assessment, alternative approaches, agri-food supply chains

## Abstract

The EU’s regulatory framework for genetically modified organisms (GMOs) was developed for “classical” transgenic GMOs, yet advancements in so-called “new genomic techniques (NGTs)” have led to implementation challenges regarding detection and identification. As traceability can complement detection and identification strategies, improvements to the existing traceability strategy for GMOs are investigated in this study. Our results are based on a comprehensive analysis of existing traceability systems for globally traded agricultural products, with a focus on soy. Alternative traceability strategies in other sectors were also analysed. One focus was on traceability strategies for products with characteristics for which there are no analytical verification methods. Examples include imports of “conflict minerals” into the EU. The so-called EU Conflict Minerals Regulation requires importers of certain raw materials to carry out due diligence in the supply chain. Due diligence regulations, such as the EU’s Conflict Minerals Regulation, can legally oblige companies to take responsibility for certain risks in their supply chains. They can also require the importer to prove the regional origin of imported goods. The insights from those alternative traceability systems are transferred to products that might contain GMOs. When applied to the issue of GMOs, we propose reversing the burden of proof: All companies importing agricultural commodities must endeavour to identify risks of unauthorised GMOs (including NGTs) in their supply chain and, where appropriate, take measures to minimise the risk to raw material imports. The publication concludes that traceability is a means to an end and serves as a prerequisite for due diligence in order to minimise the risk of GMO contamination in supply chains. The exemplary transfer of due diligence to a company in the food industry illustrates the potential benefits of mandatory due diligence, particularly for stakeholders actively managing non-GMO supply chains.

## 1. Introduction

Genetically modified organisms (GMO) may only be placed on the EU market following an authorisation procedure, and products derived from GMOs, including GM food and feed products, have to be labelled accordingly [1,2,3] (see [4] for a comprehensive overview). Compliance with these regulatory requirements is ensured by inspections (including documentation checks and laboratory tests). Importers of products that might contain GMOs have to make sure that their charges are free of unauthorised GMOs; the EU has in principle zero tolerance for unauthorised GMOs. To support compliance through the detection of authorised GMOs, a quantitative, event-specific analytical detection method is provided by the developer of the GMO in the framework of the application for market authorisation [5]. These detection methods are available for authorised GMOs as well as for those with ongoing approval procedures and allow the unique identification of the GM event.

The current regulatory framework for GMOs in the EU was developed when “classical” GMOs produced by transgenesis were state-of-the-art. However, since the entry into force of EU Directive 2001/18/EC [1], techniques were developed to produce GMOs allowing various modifications in the genome. In the EU, these methods are summarised under the term “new genomic techniques” (NGTs) and include targeted mutagenesis using CRISPR/Cas [6]. In 2018, the European Court of Justice clarified that products developed by targeted mutagenesis are considered GMOs and are subject to the same regulatory requirements as “classical” transgenic GMOs [7].

Since then, the implementation of the detection and traceability requirements for NGT products [4,8,9,10] and regulatory changes have been discussed, with the latest development being a proposal for a regulation on plants obtained using certain NGTs published by the European Commission in July 2023 [11].

Control regarding authorised GMOs and GM products covers compliance with labelling requirements, considering the thresholds for the adventitious and technically unavoidable presence of minimal amounts of GMOs in other products [2]. Control measures, including analytical control, are also important to ensure that no GMOs are present in products that are marketed as being non-GMO, i.e., conventional non-GM products and products of organic production, the latter prohibiting the use of GMOs according to Regulation (EU) 2018/848 [12]. Inspections are also essential to ensure that no unauthorised GMOs enter the EU market. Analytical methods for unauthorised GMOs, however, have to be developed by control laboratories based on publicly available information. The development of such detection methods focusses on GMOs whose presence in supply chains is suspected or known. GMOs are usually detected in the routine application of screening methods. However, the range of newly developed transgenic GMOs that are not detectable with the Matrix approach used in the EU has significantly increased in the last years [4].

In order to develop event-specific analytical detection methods for NGTs, some key information is needed. On the one hand, information on products on the market worldwide is needed, and on the other hand, product-specific information that is crucial for developing detection methods is required, e.g., (sequence) information regarding the genetic modifications present in these products. However, such information may not be readily available for different reasons: Firstly, GMOs are regulated differently worldwide, and many non-EU countries do not require GMOs to be labelled. Secondly, NGT products are not considered GMOs in several countries [13]. Thus, necessary information regarding these products is not available from regulatory authorities. The placing on the market of NGT products will increase the existing challenges regarding the detection of non-authorised GMOs, which are currently not detectable with the available screening methods but only with event-specific detection methods. Although NGT products are detectable if respective sequence information is available, the specific challenge associated with NGT products might be their unequivocal identification. It might be challenging to link the respective modification in the genome to the new genomic technique applied [9,10]. In order to solve this challenge, sufficient information on the NGT plant needs to be available [4,8,13].

Currently, no internationally consistent approach for the development and publication of detection methods exists [8]. The Biosafety Clearing House (BCH) [14] established under the Cartagena Protocol on Biosafety provides information for organisms that are subject to a national authorisation procedure. In addition, only Parties to the Cartagena Protocol participate in the BCH information exchange together with some non-Parties like the US or Canada. Against this background, an attempt to provide an overview of NGT products was made by Wageningen Food Safety Research (WFSR) and the German Federal Office of Consumer Protection and Food Safety (BVL). They have jointly developed a database that collects information relevant to GMO products (including NGT products) and their detection and identification, called “The European GMO Database” (EUginius Database) [15]. Information in the database is provided by WFST and BVL, in collaboration with several other project partners, the Austrian Agency for Health and Food Safety Ltd. (AGES), the Polish Plant Breeding and Acclimatization Institute (IHAR) and the Italian Experimental Zooprophylactic Institute of Lazio and Tuscany (IZSLT).

Globally only a few NGT plant products are already being marketed. However, several are in the pre-commercial stage, complemented by many applications in the advanced research and development stage [16]. Since an increasing number of NGT products is expected to be developed in the future for the agri-food sector, the issue of detection and traceability is urgent in order to ensure compliance with GMO regulation in the EU (for unauthorised NGTs) and maintain non-GMO production chains (for authorised NGTs in the future). In the absence of detection methods, authorities have to explore traceability strategies that are not based on analytical testing.

The traceability of food products has become increasingly important as the food and feed trade has become more global. Historically, the need for traceability arises from concerns about health and safety risks within food systems [17]. In recent years, the traceability of global food and feed supply chains has become increasingly important, also in the context of growing sustainability awareness [18].

Although current definitions of traceability differ in focus, the main principles of transparency regarding identification of origin, quantities and movements along different stages of the value chain remain at the core of the concept. Traceability, as defined by the International Organization for Standardization (ISO) and the Regulation (EC) No 178/2002 [19], involves the ability to monitor the movement of food and feed through specified stages of production, processing and distribution.

Traceability enables the effective management of the flows of goods (commodities and products) in two ways: tracing (upstream) and tracking (downstream) [17]. From the perspective of food safety, it serves as a tool for effective management, allowing for the swift removal of potentially harmful products from the market in the event of incidents. 

In the context of global value chains, the emphasis of this paper is set on the tracing aspect of imported agricultural products that might contain (unauthorised) GMOs (including NGTs). The aim of this article is to review the state of the art of traceability systems and to explore strategies to ensure that only authorised GMOs (including NGTs) enter the EU market and non-GMO production chains remain GMO (and NGT)-free.

## 2. Methodological Approach

The methodological approach followed in the present study consisted of an iterative process in which findings from literature and desk research were complemented by expert inputs from selected stakeholders via interviews and workshops (Table 1).

Initial data collection was conducted through a literature review and desktop research covering scientific literature, as well as trade data and grey literature from governmental agencies. This research and its evaluation were not limited to GMOs but also covered traceability approaches in other global value chains, taking into account approaches not based on analytical evidence. 

For the review of the status quo (Section 3), in addition to the literature review, a comparative analysis of traceability systems currently applied in different global supply chains was carried out. This analysis compared the traceability-specific requirements of different sustainability standards for global agricultural commodities. Representative standards were selected and compared using the ITC platform standardsmap.org [20]. The result of this comparison is compiled in Table 2.

On 22 December 2022, potential interviewees were contacted by email, some of them several times. Some individuals were also contacted by telephone. A total of 10 semi-structured online interviews were conducted in January and February 2023 with different types of stakeholders as listed in Table 1 (The Interview Questionnaire is available in Appendix A). The objective of the interviews was to understand current strategies applied in selected agricultural supply chains that support non-GMO labelling claims as well as to understand how conventional, non-GMO-labelled agricultural products are imported into Europe and how current monitoring of compliance with EU legislation is carried out. Interviews with experts were intended to supplement the literature and desktop research by gathering additional details on identified key topics of addressing data gaps through questions targeted at specific stakeholder groups. 

Moreover, the regulatory perspective of traceability in global value chains was assessed in a literature review. For this purpose, the general concept of due diligence and specific due-diligence-based regulatory instruments applicable to agricultural and non-agricultural global supply chains such as soy, conflict minerals and timber were examined. 

In exploring possible alternatives for the traceability of unauthorised GMOs in Europe, the elements of due diligence were examined using a specific example. The EU Conflict Minerals Regulation [21] was analysed in detail, and requirements relevant to a theoretical transfer to the field of GMOs were identified. The practical implementation was showcased by the transfer of the conflict mineral due diligence regulation to a specific hypothetical company from the food sector.

From those results, approaches for improving the traceability of products that might contain GM plants (including NGTs) were deduced. Two expert workshops with interview partners and other relevant stakeholders were conducted in April and September 2023 to validate and prioritise the identified alternative approaches to GMO traceability. The first expert workshop was held online on 26 April 2023; the second expert workshop took place on 29 September 2023 at the Federal Agency for Nature Conservation in Bonn. Hybrid participation was made possible. Thirty-four international experts from different stakeholder groups (non-governmental organisations, standard organisations, industry associations, feed trade, food producers, research, authorities, ministries) were invited to the first workshop. A total of 9 experts participated, including 6 experts from Germany and one expert each from Brazil, Ukraine and Belgium. 

Thirty-six experts from all relevant stakeholder groups were also invited to the second workshop. A total of 23 experts participated, including 19 experts from Germany, 1 expert from the Netherlands, 2 from Belgium and 1 from Ukraine. The inputs from the workshop facilitated the discussion and optimisation of the alternative traceability strategies for unauthorised GMOs in Europe as presented in Section 4.

## 3. Traceability Strategies and Their Implementation in Global Supply Chains

In order to understand the status quo of traceability practices, this section explores different existing systems and instruments. The selected instruments include voluntary sustainability standards and due-diligence-based regulations applicable to goods imported into the EU. In this section, the authors also present the case study of soy, selected due to its relevance in terms of import volumes for the European market, in further detail. The results of this analysis should contribute to understanding which traceability strategies are used in the non-GM soy supply chain imported from Brazil and how compliance with EU regulation is ensured. The instruments examined in this section provide elements for an alternative traceability strategy for GMOs not authorised in Europe as presented in Section 4. 

### 3.1. Existing Traceability Systems within Voluntary Sustainability Standards for Globally Traded Agro-Food Products

Voluntary sustainability standards (VSSs) play a pivotal role in ensuring transparency, accountability and ethical practices within global value chains. When integrated with traceability requirements, these standards can significantly enhance the management of different value chains—from primary production to market—and address increasing consumer demands regarding sustainability concerns inherent to global value chains [22].

The central concept underlying all traceability systems commonly used in voluntary sustainability standards is that the origin of a product is shown transparently and—depending on the traceability requirements—without gaps along the entire supply chain. This means that the key function of a traceability system is to collect and compile data on specific product characteristics and to trace them back along the entire value chain. This requires the establishment of a so-called chain-of-custody (CoC) system.

The CoC system and the underlying control system are key components of most sustainability standards. Both elements serve to verify the claims made through the labelling of a product, process, company or service with a corresponding sustainability standard [23]. To achieve this goal, sustainability standards must define a set of requirements, measures and means of proof that must be met along the entire supply chain. The control system, also defined and established by the standard owner, serves to verify that the stated requirements are adhered to.

Systems used to ensure traceability vary widely and are designed to be fit for purpose (e.g., could be paper-based or only go to a limited level of detail) [23]. Due to the differences in globally traded raw materials, intermediate products or finished products, each industry and standard applies different approaches to traceability.

In supply chain management, there are four main CoC models [23]:“Identity Preserved” ensures the physical separation of the certified raw material from its extraction or agricultural production to the final consumer good, allowing traceability.“Segregation” keeps certified and non-certified materials separate, allowing the mixing of compliant materials from different producers.“Mass Balance” tracks certified raw material by weight, allowing mixing before separation for accounting.“Book & Claim” involves purchasing sustainability certificates based on raw material quantity, without guaranteeing physical traceability. This model helps manufacturers meet procurement goals when direct sourcing is not possible.

Only the two CoC models “Identity Preserved” and “Segregation” can be considered for a traceability strategy for GMOs (including NGTs). The Mass Balance and Book & Claim models only provide administrative traceability. There is no physical link between the raw material used and the production supported by the certificates. These models allow producers to meet their sourcing targets if they are unable to source the required quantities of the certified product directly.

In order to understand the status quo of different approaches to traceability, the authors conducted a comparative analysis of selected recognised voluntary standards established for the certification of different agro-food products and commodities. The aim of this analysis was to identify current practices as well as initial insights for deriving alternative traceability strategies for unauthorised GMOs in Europe. Table 2 provides an overview of the aspects considered in the analysis. All of the selected VSSs are third-party-certified, requiring respective independent audits to ensure compliance with the standard criteria.

The current landscape of traceability systems within voluntary sustainability standards adhering to the “Identity Preserved” and “Segregation” models in agricultural product supply chains is characterised by the meticulous documentation and archiving of pertinent information throughout the entire continuum. In certain instances, standards incorporate online platforms or digital tools to facilitate the documentation of all actors along the supply chain. The use of blockchain-based systems for documentation is also on the horizon. Appropriately developed applications based on blockchain technology—as a new form of data storage—raise a number of expectations. It is usually pointed out that appropriately developed blockchain applications can facilitate the work of documentation required for traceability and minimise the risk of fraud in documentation. However, a research team led by Commandré was unable to confirm the fulfilment of these expectations of blockchain technology based on their study results [24].

The operational framework for documentation in the standards analysed includes the following key aspects:At the outset, the farmer is obliged to meticulously maintain purchase receipts for seeds or relevant inputs, alongside comprehensive records of yields and resale transactions. In most cases, products are endowed with a unique identification number right at the farm level.Subsequently, the initial processor or intermediary entity is tasked with documenting vital details, such as the quantity and origin of certified raw materials received. Moreover, in cases of resale, pertinent information including the buyer’s name and address, loading or dispatch/delivery dates, document issuance dates, certificate number, product description and delivered quantity has to be recorded, along with all associated transport documents.The verification process is conducted by external auditors hailing from accredited certification bodies who conduct on-site assessments. These audits encompass a comprehensive review of documents in conjunction with physical site visits including plausibility checks, e.g., to ensure that the seed input is consistent with the crop yield. The frequency of the audits and their intervals are specified in the respective standards. Often, standards employ a risk-based approach, wherein regions with an elevated risk of misdeclaration necessitate a higher frequency of audits, as opposed to regions deemed to have a lower risk of misdeclaration.

### 3.2. Status Quo of Labelling and Traceability of GM Crop Imports to Europe

In the complex web of global trade for agro-food products, the traceability of GMOs has emerged as a critical concern. Ensuring the accurate identification and tracking of GMOs within international supply chains is not only pivotal for regulatory compliance but also fundamental for meeting consumer demands for transparency. This section dives into the current landscape of labelling and traceability for the import of GM crops (and derived products) to Europe.

Current traceability practices for agricultural products containing GMOs rely on the data shared by actors from the first stages of the supply chain. When commodities produced through genetic modification techniques move along the value chain, specialised business-to-business (B2B) markers are essential. These markers, also known as unique identifiers, serve to track the specific GMO lineage but are not communicated to the end consumer. Instead, the obligation to furnish this information extends solely to the subsequent actor in the supply chain. At the same time, whether data indicating the presence of GMOs are shared with the next tier in the supply chain is determined both by the regulation in the country of origin of these commodities and the regulation in the importing country. If the national regulation of the country of origin does not require GMO labelling, the obligation to report data about the corresponding B2B marker will be determined only by the importing country. 

In this context, importers into the EU are required to rely on the information received from their suppliers while ensuring compliance with EU GMO labelling requirements. Conventionally, these data are transmitted through the delivery note, without necessitating documentation of the goods’ origin, cultivation details or primary producer information.

It is crucial to note that the challenge of traceability transcends GMOs alone. It is, e.g., also of paramount importance in substantiating compliance with various criteria of sustainability standards (see Section 3.1). This encompasses an array of requisites, aimed at ensuring compliance with tangible but also intangible attributes. Common intangible attributes for which standards must apply traceability are claims about fair trade or production without child labour. Other common intangible areas of application for traceability in standards include verifying that agro-food commodities originate from supply chains free of deforestation or are sourced from organic and fair cultivation systems. Additionally, traceability serves as a cornerstone for certifying non-GMO agricultural supply chains, further underscoring its broad-reaching significance in contemporary agricultural trade practices.

### 3.3. Traceability of Non-GMO-Labelled Globally Traded Commodities: The Case of Soy from Brazil

The top four GM crops in the world are soybean, corn, cotton and canola [25]. The case of soy, which is used in a wide range of food and feed products, has become a key issue in the international trade in agricultural raw materials. Soybeans account for 50% of the world’s GMO cultivated area [25]. A substantial portion of the traded volumes originate from regions with a significant use of GMOs in agricultural practices. According to a 2021 report from the Global Agricultural Information Network (GAIN), the share of GMO varieties currently used for soy cultivation in selected countries ranges from 50–65% for Ukraine to 100% for Argentina. For Brazil, the share of GMO varieties used in soy production is 96% [26]. These data indicate that in the main soy-growing countries, almost exclusively GM varieties are cultivated. By cross-checking these findings with available data from international trade databases [27], it becomes evident that imports of soy and its byproducts (oil, meal, oil cake) account for the largest share of agricultural commodity imports to the EU in terms of volume. According to the UN’s COMTRADE database, imports of soybeans and their byproducts into the EU totalled around 31 million tonnes in 2021. Most of the soy imported into Europe is genetically modified and used as animal feed. Along with rapeseed meal, soy meal is one of the most important protein feeds. A closer look at the country of origin shows that the largest share is imported from Brazil [25]. Despite the high proportion of GM soy grown (96%), Brazil is still one of the few countries that exports large quantities of non-GM soy. Against this background, the import of non-GM soy from Brazil to Europe was selected as a case study to analyse the traceability strategies implemented for these import flows. 

The case study provides an overview of the detailed practices involved in the traceability of non-GM soy produced in Brazil and its chain of custody to the point of entry into the Europe to ensure compliance with current regulations. Insights into the traceability practices described in the case study are based on interviews with a Norwegian company that exclusively imports non-GM soy and whose entire supply chain is also certified to the ProTerra standard [28] (see Table 2). 

The company’s main supplier is based in Brazil. The soybeans are grown in the Mato Grosso region, a key location for soy production in Brazil. All actors in the supply chain follow a strict process to ensure the segregation of non-GM soybeans at all stages of production, storage and transport. Non-GM soy is only harvested using machinery specifically designed for this purpose or machinery that has been thoroughly cleaned to avoid contamination with GM soy. Once harvested, non-GM soybeans are stored separately in special silos. These silos are then transported to Porto Velho for storage before being shipped down the Madeira River to the port of Itacoatiara, where they are stored in warehouses dedicated to non-GM soybeans. From there, the soybeans are shipped by sea (see Figure 1).

All relevant stages in the value chain are documented. This means that the farmer has to document the purchase of non-GMO seed (e.g., with the relevant purchase receipts). It is double-checked whether the receipts for non-GM seeds match up with the actual harvest. The certified crop is given a batch number, which is passed down the supply chain. Silos are clearly marked and sealed for extra security. The quantities and origins of the certified raw materials received and their processing must then be documented throughout the supply chain (e.g., as follows: name and address of the buyer, name and address of the seller, date of loading or dispatch/delivery, date of issue of documents, certificate number, quantity of product delivered, relevant transport documents).

Furthermore, samples are taken for GMO testing using both rapid tests and PCR tests right from the early sowing stage through to each storage phase. This means seeds are also tested for the presence of GMOs. Accredited and independent laboratories handle these tests. Only shipments with negative test results move on to the next stage, with all cargoes held in quarantine until they receive the green light based on the test results. All the test results (including date and time) are carefully logged in a database run by the independent laboratory, giving a detailed record of responsibilities and shipments. The company also keeps a parallel paper record of the entire supply chain.

The interviewed Norwegian company stated that every batch of soy arriving in its port goes through about 3800 GMO tests, including quick tests. The company also regularly trains and inspects its suppliers to stress the importance of following the rules. When a ship with soybeans docks, the company promptly informs the relevant food authorities. This means that when the soybeans are unloaded, the authorities take samples right there at the port and test them for GMO content.

It is crucial to highlight that in order to be sure about the GMO status of a product, one needs to have separate supply chains for non-GM and GM products. Only machines, like harvesters and tractors, exclusively used for non-GM products can be used for the harvest and transport to the storage silos. In countries like Brazil, this is mostly doable on large industrial farms. The basic idea of a non-GM soy supply chain is the same for all traceability systems of agricultural products that follow the “Identity Preserved” and the “Segregation” model, even if there are small differences in the specifics. Using one of these two models is necessary for non-GM goods when attempting to comply with EU regulations.

The efforts described above for the segregation of non-GMO and GMO-containing raw materials and intermediate products result in additional costs. This point was raised in the interviews conducted. Many respondents were unable to quantify the costs of ensuring non-GM supply chains. This was generally due to the fact that most respondents did not market both GMO and non-GM products. The standard organisations interviewed claimed that the cost of certification is minimal compared to the cost of segregation. However, certification gives the legitimacy to claim a premium in the European market. One German food retailer stated that the prices for globally traded certified non-GM soybean meal are about 17% higher than those for conventional soybean meal. Another interviewee from a food producers’ association explained that, in general, there are no major price differences between non-GM soy from European value chains and globally traded non-GM soy. Prices are very volatile and depend on various factors, such as weather-related fluctuations in crop yields. 

The additional costs of non-GM supply chains were also included in the literature review, and two different studies were found and analysed. In Brazil, the cost of non-GM soy includes several variables, including the availability of non-GM seeds; the need to establish a 20-metre buffer zone to prevent cross-pollination; increased maintenance costs to control weed growth; and the aforementioned efforts to avoid contamination during harvest, storage and transport [29]. 

In a study published in 2014, the additional costs of production and supply chain segregation in Brazil and the premium surcharge for non-GM soy were calculated and estimated to be around 21%. Just over half of the additional costs (around 51%) are borne by the farmer. Just under half (about 49%) of the additional costs are due to separate storage and transport [30].

### 3.4. Due-Diligence-Based Regulatory Instruments Contributing to Traceability of Imported Goods

The concept of supply chain due diligence and its relevance for the traceability of imported goods has its roots in the UN Guiding Principles on Business and Human Rights which were published in 2011 at the end of the so-called “Ruggie process” [31,32]. Those principles assigned companies a central role in respecting human rights worldwide. Until then, the question of the human rights responsibility of private companies was at the discretion of nation-states, which can pose challenges in their enforcement [31]. The UN Guiding Principles thus for the first time assigned their respective roles to both states and globally operating businesses. In the following years, the due diligence concept, developed in the context of human rights, has been increasingly expanded to include, depending on the standard or legislation, impacts on the environment as well as society in a broader sense [33,34].

In recent years, a number of binding legal provisions based on the concept of human rights due diligence or incorporating some of its elements have been enacted. Those regulations include comprehensive regulatory approaches like the German Supply Chain Act (Lieferkettensorgfaltspflichtengesetz—LkSG) [35] as well as sector-specific solutions such as the EU Conflict Minerals Regulation [21] and the EU Regulation on deforestation-free products [36]. Finally, the European Commission has published a legislative proposal for a comprehensive due diligence regulation (Corporate Sustainability Due Diligence Directive (CSDDD)) [37]. Looking at the above-mentioned regulations, valuable insights were drawn regarding their transferability to alternative traceability strategies to prevent the import of unauthorised GMOs in the EU.

Due diligence defines companies´ responsibility for human rights violations and adverse environmental impacts even when they are not directly committed by the company but by business partners [38]. This means that the respective responsibilities apply to the entire value chain of a company and its products. So, if legislation is in place, the due diligence requirements apply to companies that are headquartered in or import into the regulated area (here, the EU), but they also indirectly affect companies in the supply chain that may be located outside the regulated area. 

Critics often argue that it is impossible for businesses to meet such a comprehensive responsibility, especially if, for example, it is a distant supplier that has caused the negative impacts [39,40]. This argument, however, is based on a misconception. Firstly, due diligence differentiates with regard to the degree of responsibility according to the level of actual “involvement”. For example, companies that directly cause a human rights violation should stop doing so, avoid doing so in the future and also provide remediation. If companies are merely “linked” to the human rights violation, they are only required to use their influence on the company that has caused the damage [38,41,42]. Secondly, due diligence is a risk-based approach. It does not establish an “obligation of result” but an “obligation of means” or “reasonable care”, meaning that companies have to first identify their risks along the value chain. Subsequently, they must take measures to prevent, avoid and mitigate those risks and, in case harm has already been caused, provide remedy. However, companies do not have to guarantee that no negative impacts occur [38,42]. The concrete solutions resulting from the regulations under consideration are explained in more detail in the following section.

## 4. Alternative Traceability Strategies to Prevent the Import of Unauthorised GMOs into the EU

### 4.1. A Risk-Based Approach to Traceability to Prevent the Import of Unauthorised GMOs into the EU—Using Elements of Due Diligence Legal Obligations

Even though a policy of zero tolerance currently applies to the import of non-authorized GMOs (including trace amounts) into the EU, apart from lab-based testing, no other methods with which companies have to comply are specified. Our exercise is attempting to fill this gap. Focussing on a hypothetical due diligence system for GMOs, importers of the relevant agricultural products would have to set up a traceability system allowing them to provide information on the origin of their goods and whether or not they contain GMOs. Such a traceability system would have to include documentation on the origin and breeding methods of the seeds used, as well as on each processing and transport step in the supply chain. It would also need to document the mixing of different batches.

When it comes to human rights and environmental due diligence, obligations generally place responsibility on the individual company. However, for companies to successfully fulfil their due diligence obligations in global value chains, the interaction of a large number of different actors as well as the implementation of various instruments is necessary. Usually, companies do not themselves control or audit every single supplier but rely on certification and audit schemes.

For many sectors and products, a variety of voluntary certification schemes and standards already exist. Following a content and process alignment with the regulatory requirements, these voluntary schemes and standards can be recognised by regulators and be used as proof of due diligence compliance by the firms.

Often, such tracking and tracing systems certify so-called intermediaries, such as smelters in the case of the Conflict Minerals Regulation. Similar actors could be identified for trade in agricultural products (in the case of soy, e.g., oil mill operators).

Most soy importers have to implement such a tracing system that allows them to identify suppliers and countries where the respective commodity has been cultivated anyway in order to be compliant with the EU Regulation on deforestation-free products. The only data missing concern the cultivated plant variety. This would mean the labelling of GMOs (as well as food and feed produced from GMOs) and the unique identifier according to Regulation (EC) 1830/2003 [3]. Importers of soy would also have to be transparent about which kind of breeding technologies have been applied with regard to the seed used for the cultivation of the imported soy.

Important information for assessing the risk of unlabelled or unauthorised GMO imports or GMO contaminations comprises information on GMOs available on the market worldwide. Ideally, this information is compiled in an international database and comprises the name of the event, modified plant species, modified variety, producer, countries with authorisation (in case authorisation is necessary), countries with actual market availability and countries with cultivation and/or field trials (possibly with information regarding the regions).

For the traceability of GMOs, or as a basis for risk-based controls on unlabelled or unauthorised GMO imports, further information could be helpful and collected at the EU level. This could include an overview of the main countries exporting plant products into the EU for which GM varieties are available worldwide and an overview of the main import points into the EU (e.g., harbours). 

Additionally, a grievance mechanism should be established, either by the supervisory body or by the companies. The benefit of such a grievance mechanism is that suspicious cases can be reported, for example by civil society actors. Thereby, an additional source for risk management is generated.

A corresponding GMO due diligence regulation would also open up the possibility of sanctioning violations, for example through fines, the use of blacklists or even corporate liability. 

### 4.2. Validation of the Developed Alternative Risk-Based Approach to Prevent the Import of Unauthorised GMOs into the EU

The project team presented the developed approach for an alternative traceability strategy for discussion in an expert workshop. The aim was to validate the developed approach and to adapt it if necessary.

When proposing due diligence, the additional burden for companies is not intuitively obvious. To illustrate what a company might expect given a hypothetical GMO due diligence regulation, the authors performed a hypothetical exercise as described below and visualised in Figure 2.

In order to estimate what additional measures companies would have to take if a hitherto hypothetical due diligence regulation for unauthorised GMOs were to be adopted, the EU Conflict Minerals Regulation was analysed as a first step. Those requirements relevant to a theoretical transfer of the due diligence approach to GMOs in agricultural supply chains were identified (see Table 3). The authors chose this regulation because exemplary companies have well documented how they meet the due diligence requirements of this regulation. Such well-documented practical examples were essential to work out what companies should expect. 

Table 3 provides an overview of all Articles of the EU Conflict Minerals Regulation [21] and points out their relevance to the GMO practice example. Although many requirements would certainly be needed if a due diligence regulation for GMOs in agricultural supply chains were to be introduced, many of them were found to be too detailed for the intended theoretical transfer at this stage. Nevertheless, requirements from Articles 1, 4, 5, 8 and 14 (marked with * in Table 3) were found to provide good references regarding what due diligence could theoretically look like for a practice example in the food industry. 

Based on an analysis of the EU Conflict Minerals Regulation, the authors have developed a model to exemplarily demonstrate the application of a risk-based approach in practice. This was accomplished through scenario creation to describe what the introduction of due diligence elements could mean for a fictional company in the food industry.

This fictional company with the name “FoodAlternatives” produces various products in the food sector and has its headquarters in Germany. It produces a broad variety of products, comprising various agricultural goods. The agricultural goods are imported from all over the world. The company is a member of a non-GMO association and has labelled its products as non-GMO. 

In our exercise, a hypothetical new EU regulation makes due diligence efforts for GMOs in agro-food supply chains mandatory. This means that all companies would have to ensure that no unauthorised GMOs enter the supply chain. The company “FoodAlternatives” is looking for guidance on how to implement the new requirements. With reference to the relevant articles identified, the authors have described, in Table 4, what the company needs to do in order to comply with the regulation.

The exercise reveals that many important due diligence requirements are already being met by companies selling certified non-GM products. It can be assumed that the key elements of a hypothetical due diligence regulation for GMOs are already being implemented by a company selling non-GMO products based on imported ingredients.

## 5. Discussion

Focussing on GMOs, it is important to note that traceability requirements exist in the EU for authorised GMOs. Additionally, the non-GM sector currently employs traceability, supported by analytical controls to ensure non-GMO status. Analytical controls in the GMO sector, including NGTs, would be the optimal solution to guarantee non-GMO status and ensure that only authorised GMOs enter the EU market. The challenge is that the required detection methods or the appropriate information for developing such detection methods might not be readily available. Here, the existing international GMO databases could play a central role in providing access to information concerning detection and identification. Consideration should be given to supplementing these databases with additional information on the type of GMO and data to help assess the risk that a particular supply chain may contain unauthorised GMOs. Another option would be to create a completely new database that hosts information to facilitate the detection and identification of GMOs including NGT products. Ideally, such databases should provide access to DNA sequence information regarding the respective genetic modification(s) present in a particular GMO and should be retrievable in automated ways. As the analytical detection and identification of NGTs will likely be challenging for some products even with an improved information base, an alternative traceability strategy needs to be developed in addition.

Today, there are many products on the European food market that have claims that are not discernible by visible product characteristics (e.g., grown without the use of various pesticides, grown on land that does not contribute to further deforestation, grown and harvested without child labour, European certificate of origin, non-GMO, etc.). Not all existing claims can be substantiated by analytical methods. Our analysis has shown that there are traceability strategies that ensure the postulated properties, even if they cannot be proven analytically. The central concept underlying all traceability systems is that the ingredients of a product can be traced back through the whole supply chain up to the agricultural production or even to the origin of the seed material used to grow the respective crops.

Given the current GMO legislation in Europe, it is theoretically possible to implement full traceability of agricultural products that comply with existing EU legislation. In practice, there are now a large number of traceability systems for globally traded agricultural products that are certified to meet certain sustainability requirements. However, these systems are voluntary. The information required for traceability is provided voluntarily by actors throughout the supply chain. Typically, operators benefit by obtaining a premium price for their product on the European market. However, operators marketing conventionally produced products or NGT products may have no interest and no added value in providing the relevant batch traceability information. For those operators that would have to provide the necessary data for traceability, there is no reason or motivation to do so. However, this means that importers into the EU have to be legally obliged to prove that they only import approved GMOs. In the event of false declarations, importers would face severe sanctions.

The traceability systems currently in use in different sustainability standards are voluntary and do not in themselves provide a solution for the traceability of GMO products under current European GMO legislation, because the implementation of traceability is not defined. However, the transfer into an appropriate legal regulation could be an important step for an effective traceability strategy. 

It must be emphasised that traceability itself is not the regulatory objective. It is merely a means to an end, a prerequisite for fulfilling the described due diligence obligations in place or a hypothetical due diligence regulation for GMOs. Only when a supply chain is known can risks be assessed and, if necessary, addressed. 

The risk-based perspective of due diligence would have additional implications. Above all, it means that the tracing system should take into account different risk levels. This means, for example, that depending on the product or the country of origin, the extent of the controls would differ. Accordingly, the regulatory bodies or supervisory authorities should develop respective risk categories. On the one hand, these categories would be for their own use so they can, e.g., focus document checks or laboratory tests on high-risk cases; on the other hand, this information would be made available to other companies (e.g., in an international database) so that risk categorisations can be harmonised.

The exemplary transfer of due diligence to a company in the food industry shows that many important due diligence requirements are already met by selling certified non-GM products. The main elements that can be assumed to have already (partly) been introduced in the context of non-GMO certification comprise (1) a policy on how to avoid GMOs in supply chains, (2) an overview of agricultural supply chains and actors, (3) a traceability system based on the respective information about supply chains and involved actors, (4) the use of third-party audits to ensure compliance and minimise risks and (5) a management position with responsibility for the tasks described before.

Selling certified non-GM products is very likely to even go beyond compliance with a due diligence regulation, as it excludes all GMOs, including those authorised in the EU. 

In this context, it should be considered that the existing concerns of stakeholders currently implementing non-GMO supply chains may be unfounded. On the contrary, a mandatory introduction of due diligence might give certain companies advantages over players who have not yet been actively investigating and managing their supply chains with regard to GMOs so far. 

## 6. Conclusions

Undoubtedly, for some NGT products, challenges exist regarding detection and especially unequivocal identification. For this reason, an adaptation of the current GMO traceability system might be necessary. However, the argument that European GM legislation needs to be amended because of the existing challenges regarding analytical control methods is not entirely valid. The existing challenges might be compensated by other means since there are a number of regulations that prohibit the import of certain products without analytical control methods available to prove that the imported products comply with the regulation. Notable examples are the EU Conflict Minerals Regulation or the new Regulation on deforestation-free products. Consequently, there is no need to reduce GMO regulatory standards for NGTs just because detection of them poses a challenge.

Regarding improvements in traceability within the non-GMO sector, the concept of the “reversal of the burden of proof” is relevant. It necessitates the declaration of not only authorised GMOs (as per current requirements) but also the absence of GMOs. This declaration would have to apply to everyone, not just the non-GM sector. This could be ensured through a specific due diligence regulation that requires that importers introduce a traceability system for their supply chains and that importers identify and assess the risk of GMOs in their supply chain. This risk assessment in turn would require a range of information on GMOs, such as a regularly updated overview of the main countries exporting plant products to the EU for which GM varieties are available worldwide and an overview of the main points of entry into the EU (e.g., ports). Preferably, these data should be made available in a robust international database that also contains easily retrievable DNA sequence information on GMOs. This information would enable importers of agricultural products to conduct the required non-analytical assessment of risks regarding GMO contamination.

Overall, there is an urgent need for more transparency in our global agricultural supply chains, not only with regard to GMOs. Against the backdrop of the global climate crisis and the dramatic loss of biodiversity, the European Parliament and the Council of the European Union adopted the EU Regulation on deforestation-free products (EU) 2023/1115 in May 2023. 

This regulation now needs to be implemented by European importers. It obliges them to disclose their supply chains and assess them for risks related to deforestation. As soy is one of the agricultural products with a high risk of deforestation, there are synergies with the issue of GMOs. In this context, better traceability alternatives applied to globally traded agricultural products could also contribute to ensuring that only authorised GMOs are imported to the European Union.

## Figures and Tables

**Figure 1 foods-13-00369-f001:**
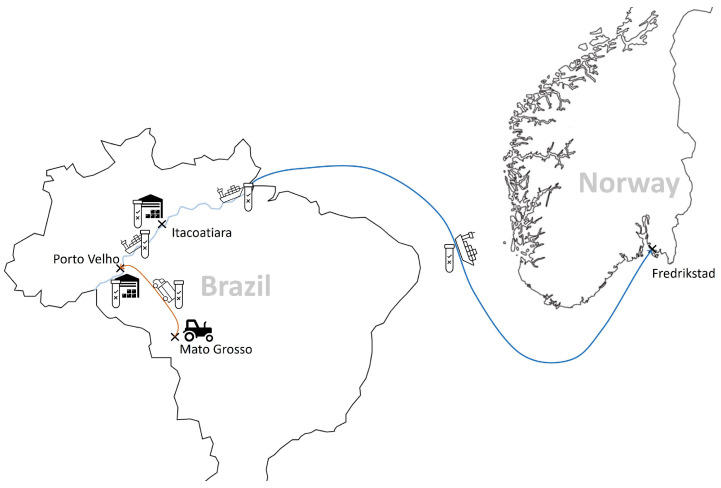
The chain of custody of certified non-GM soy from Brazil imported into Europe (source: © Öko-Institut e.V. 2023).

**Figure 2 foods-13-00369-f002:**
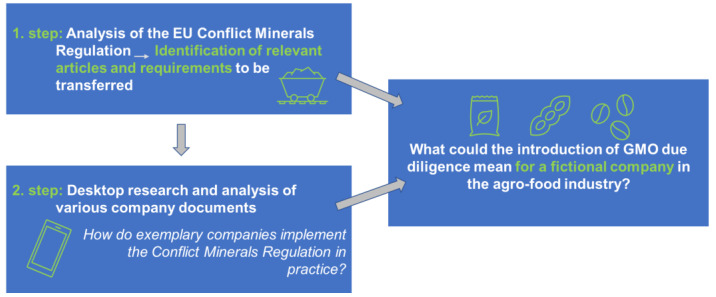
Validation process for understanding the practical implications of the developed alternative traceability strategy to prevent the import of unauthorised GMOs into the EU (source: © Öko-Institut e.V. 2023).

**Table 1 foods-13-00369-t001:** Type of stakeholders and number of interviews conducted.

Stakeholder Type	Number of Interviews
European producers of soy-based intermediates for the food and feed industry	1
Associations (food producers, processing industry)	3
NGO	1
Science/research	1
Voluntary sustainability standard organisations	2
Authorities	1
Agricultural trade company	1
European food manufacturers *	0

* Note that four European food manufacturers were contacted, but none participated in the study.

**Table 2 foods-13-00369-t002:** Comparative analysis of traceability systems in selected voluntary sustainability standards for globally traded agro-food products.

	Better Cotton Initiative (BCI)	Bonsucro	Fair Trade USA (APS for Large Farms and Facilities)	Rainforest Alliance—2020	Roundtable on Sustainable Palm Oil (RSPO)	Union for Ethical Biotrade (UEBT)	ProTerra Europe	Round Table on Responsible Soy Association—RTRS	Donau Soja Standard
**Product/** **commodity**	Cotton, fibres	Sugarcane	Cocoa, coffee, sugarcane, tea	Various (including cocoa, coffee, tea, bananas, palm oil)	Palm oil	Various (including honey, flowers, nuts, spices, sugar, tea)	Raw materials, ingredients or multi-ingredient products of food and feed including soybeans, sugarcane	Soybeans	Soybeans
**CoC** **models** **used**	Segregation (farm to ginner) Mass Balance (after ginner level)	Mass balance	Segregation Mass Balance (only allowed for cacao, sugar, tea and fruit juice)	Segregation Mass Balance (only possible for flowers, processed fruits and coconut oil)	Identity Preserved Segregation Mass Balance		Identity Preserved/Segregation (or combination of both)	Segregation Mass Balance (“conventional” or “Country Material Balance”) Book & Claim	Identity Preserved
**Dedicated CoC guidelines/** **annex**	Yes	Yes	No	Yes	No	No	No	No	Yes
**Non-GMO criteria**	No	No	No	No	No	No	Yes, core principle	Yes (additional standard)	Yes, core principle, only non-GMO varieties listed in the EU Variety Catalogue or respective national catalogues allowed
**Input material/** **seed** **traceability criteria**	Yes, seed must be traced back to BCI ginnery			Yes, origin matching requirements for cocoa beans and nibs (regional approach possible with few exceptions)		Yes, at least identifying the country of cultivation or wild collection	Yes	Yes, proof of the use of non-GMO seeds (documentation of the entire seed purchase)	Yes, verification of origin based on analytical results. Plausibility check based on risk-based approach
**Documentation requirements**	All invoices and shipping documents			Purchase and sales documents linked to physical deliveries of certified, multi-certified and non-certified products. All transactions are recorded	All receipts of RSPO-certified fresh fruit bunches (FFBs) and deliveries of RSPO-certified crude palm oil and palm kernel on a real-time basis		Crop type and volume records (including seed, planted area and plots). Analysis reports. Process, storage, shipment, reception, processing records		All delivery notes and invoices (including lot number and code of certification body)
**Sampling on representative parts of the operation**	Yes	Yes		Yes	Yes		Yes		
**Product testing requirements**	No	No		No	No		Yes (Immunologically based screening using strip tests and PCR analyses)	Yes, PCR tests at various stages of the value chain (producers, first buyers of the harvest)	Yes, rapid tests, PCR tests and risk-based origin analysis using the Soy Isotope Database
**Identification/** **labelling** **requirement**	Unique physical identification through batch numbers, labelling or visual and physical identification throughout the supply chain		Physical and visual identification during processing (through lot numbers, record keeping, etc.)		Unique identification number available in all documentation for incoming and outgoing product quantities		Lot numbers linked to traceability information, physical labelling of facilities and conveyances. Unique identification number of received consignments		Lot and quality label “Donau Soja”
**Digital tools to support traceability**	Better Cotton Platform (BCP) for electronic documentation			Online traceability platform	Platform PalmTrace			RTRS Trading Platform	Donau Soya IT database IT-supported batch certification system

**Table 3 foods-13-00369-t003:** Overview of the structure and content of the EU Conflict Minerals Regulation.

Article in EU Conflict Minerals Regulation	Content and Relevance for GMO Practice Example
Article 1 *	**Subject matter and scope**—Comprises a clarification of the regulation´s scope and introduction of threshold values for its applicability. Defining a clear scope and introducing threshold values need to be considered for a theoretical transfer to the GMO sector.
Article 2	**Definitions**—Explains how specific terms used in the regulation shall be understood. Although this is highly relevant for all (real) regulations, it does not add any value at the stage of a theoretical transfer to the GMO sector.
Article 3	**Compliance of Union importers with supply chain due diligence obligations**—Requests that union importers have to comply with the regulation, assigns responsibility for checks to the competent authorities and introduces the possibility that due diligence schemes might apply for recognition by the European Commission. As the requirements are elaborated in more detail in other Articles, no transfer for the theoretical practice example in the GMO sector is necessary.
Article 4 *	**Management system obligations**—Describes, on a more abstract level, which management systems need to be introduced by importers of conflict minerals into the EU. Some of the rather holistic requirements help to understand the theoretical transfer to the GMO sector.
Article 5 *	**Risk management obligations**—Comprises a more detailed description of how risk identification and assessment should be conducted. A transfer to the GMO example is very important to understand what due diligence efforts could look like in the sector.
Article 6	**Third-party audit obligations**—Specifies how third-party audits should be carried out. A theoretical transfer to the food sector is not necessary as there are well-established routines that could be used for this purpose.
Article 7	**Disclosure obligations**—In addition to rather general disclosure requirements, it is specified in more detail how companies must share certain information. As public reporting is also requested in Article 4, a transfer to the GMO sector does not add any additional value.
Article 8 *	**Recognition of supply chain due diligence schemes**—Introduces the possibility of due diligence schemes becoming officially recognised by the European Commission. If schemes are successfully recognised, members are automatically considered compliant. Found to be relevant for a theoretical transfer, as this might be applicable for (existing) non-GMO certifications.
Article 9	**List of global responsible smelters and refiners**—Description of what the European Commission shall do to provide a list of global responsible smelters and refiners. Relevant for the practical GMO example in order to analyse if there are comparable mechanisms that support companies with the implementation of due diligence.
Article 10	**Member state competent authorities**—Requirements for how responsible competent authorities shall be assigned with the application for the regulation. These organisational requirements are found to be too detailed for the intended transfer to the GMO sector.
Article 11	**Ex post checks on Union importers**—Outlines how ex post compliance checks shall be carried out. These requirements are found to be too detailed for the intended transfer for the GMO sector.
Article 12	**Records of ex post checks on Union importers**—Establishes rules on how ex post checks shall be documented. These requirements are found to be too detailed for the intended transfer for the GMO sector.
Article 13	**Cooperation and information exchange**—Depicts how competent authorities of Member States shall cooperate and exchange information. These requirements are found to be too detailed for the intended transfer for the GMO sector.
Article 14 *	**Guidelines**—Defines that the European Commission will provide (non-binding) guidelines to support implementation of the regulation by economic operators. This shall include a regularly updated list of conflict-affected and high-risk areas. Relevant for the theoretical practice example, as guidelines for risk evaluation could also facilitate applicability here.
Article 15	**Committee procedures**—Description of an assistant committee. These requirements are found to be too detailed for the intended transfer for the GMO sector.
Article 16	**Rules applicable to infringement**—Requirements on how to follow up on non-compliance. These requirements are found to be too detailed for the intended transfer for the GMO sector.
Article 17	**Reporting and review**—Defines how Member States shall report back to the European Commission. These requirements are found to be too detailed for the intended transfer for the GMO sector.
Article 18	**Methodology for calculation of thresholds**—Elaborates how threshold values shall be calculated. These requirements are found to be too detailed for the intended transfer for the GMO sector.
Article 19	**Exercise of the delegation**—Defines under which conditions delegated acts might be adopted. These requirements are found to be too detailed for the intended transfer for the GMO sector.
Article 20	**Entry into force and date of application**—Specifies when different sections of the regulation will enter into force. These requirements are found to not deliver any additional value for the intended transfer for the GMO sector.

* Indicate relevant individual Articles for a theoretical transfer to the GMO sector.

**Table 4 foods-13-00369-t004:** Exemplary scenario for practical implications of mandatory due diligence for GMOs in agro-food supply chains.

**Article 1—Subject matter and scope**	** *Practical implications* **
The regulation applies to Union importers of agricultural food and feed materials (raw and pre-processed products). The regulation does not apply to importers below certain *(to be defined)* threshold values.	FoodAlternatives needs to conduct due diligence for all agricultural products used in its products. Only those agricultural products bought and used in minor quantities below certain threshold values can be exempted
**Article 4—Management system obligations**	** *Practical implications* **
**(4a)** Respective importers shall adopt and communicate to suppliers and the public their supply chain policy for their purchased agricultural raw materials.	Prior to the new regulation, FoodAlternatives had published on its website its reasoning and strategy for why and how it was able to obtain its non-GMO products. With the introduction of the new regulation, the company complements the reporting. It now includes various references to the EU regulation and sets out how the different requirements are implemented in practice. The company´s suppliers are already well aware of their non-GMO policy and do not need extra communication.
**(4b)** Importers shall introduce due diligence consistent with OECD due diligence guideline Annex II.	The company reviews the regulation and recognises that it fulfils almost all of the criteria mentioned. Only minor adaptations are necessary—a detailed description of their due diligence activities can be found below under Article 5.
**(4c)** Importers shall assign senior management responsibility to supply chain due diligence.	The company has a senior manager responsible for supply chain management. Her previous responsibilities included setting up appropriate non-GMO supply chains and overseeing the certification process. She is now also responsible for overseeing the due diligence process.
**(4d)** Importers shall incorporate due diligence requirements in contracts and agreements with suppliers (in line with OECD guidance).	The company only buys non-GMO agricultural goods, and its contracts oblige suppliers to deliver only goods with GMO contamination below the current EU threshold. There is no urgent need to adapt current contracts.
**(4e)** Importers shall establish a grievance mechanism, individually or in collaboration with other operators or by facilitating resources for external experts.	A contact form is provided for anonymous reporting of violations. Incoming reports are reviewed on a regular basis and follow-up decisions are made. *One of the first reports comes from an environmental NGO in the US, which found a new variety of maize produced using an NGT that has been developed and is already sold on the national market. Based on this information, the company decides to investigate its own maize supply chain in the US in the context of its due diligence efforts.*
**(4f)** Importers shall introduce a traceability system for the supply chain. Therefore, they shall document information on the imported food ingredient´s name, name and address of the suppliers, country of origin and imported quantities.	FoodAlternatives has built up non-GMO supply chains over various years. For this purpose, different models are used. For its sugar cane supply, the company directly works together with a cooperative in Colombia. Therefore, the company has an overview of the whole supply chain and all involved actors.
**(4g)** Imported food ingredients must have proof that they do not contain GMOs that are not authorised in the EU. Records of third-party audits have to be provided. If no audits are available, further information on the supply chain has to be shared.	Soy is mainly bought from Brazil. The company has identified and established trade relations with a supplier that offers non-GMO-certified soy. As the used third-party non-GMO certification is accepted under the new due diligence regulation (see Article 8), the company does not need to investigate this soy supply chain itself. Also, for other supply chains that are third-party-certified with a recognised scheme, detailed traceability data are available at the supplier level. In these cases, there is no need for the company to collect data for itself.
**Article 5—Risk management obligations**	** *Practical implications* **
**(5.1a)** Importers shall identify and assess the risk of adverse impacts of their supply chain, based on the information collected (see Article 4).	The established non-GMO-certified Brazilian soy supply chain and the non-GMO-certified Colombian sugar cane supply chain are found to not represent increased risks for GMO contamination. However, based on the NGO report (see above), the maize supply chain originating in the USA represents an increased risk. The investigation of the supply chain and especially the involved seed producers does not confirm the possible use of an NGT maize variety. But it reveals an (unintended) contamination risk, as one intermediary trades with GMO and non-GMO maize and does not have strictly separated transport fleets. Therefore, FoodAlternatives decides to investigate further within this supply chain. Systematic laboratory testing over the next months shows regular contamination with known GMO varieties above the EU threshold.
**(5.1b)** Importers shall implement a strategy to respond to identified risks in line with OECD due diligence guidance. **(5.2)** In this context, mitigating risks does not automatically mean stopping trade in respective regions, but trade might be continued while risk mitigation efforts are implemented at the same time. If trade is continued (or only temporarily suspended), a risk mitigation strategy has to be developed together with concerned stakeholders (government authorities, civil society organisations, affected third parties, etc.).	Based on the identified source of contamination, FoodAlternatives decides to restructure its maize supply chain from the USA. The company is able to identify a supplier that established a completely separate supply chain, including all intermediaries and transport companies. Also, the used seed varieties are disclosed and do not include NGT maize.
**(5.4)** Where available, existing third-party audits might be used as part of the risk mitigation strategy.	FoodAlternatives already uses various raw materials officially certified as non-GMO. Many of them use a certification scheme that is officially recognised by the European Commission (see Article 8). For the respective supply chains, the new regulation does not have any implications for trade relations and documentation requests. As FoodAlternatives is a member of a non-GMO association, it is also able to make use of regularly updated risk evaluations for supply chains from specific countries. The latest risk update confirms a low risk for sugar cane sourced from a specific region in Colombia. As this is in line with the risk evaluation conducted by the European Commission (see Article 9), no third-party audit is necessary.
**Article 8—Recognition of supply chain due diligence schemes**	** *Practical implications:* **
**(8.1)** “Scheme owners” can apply to the European Commission to have their due diligence scheme recognised.	FoodAlternatives buys certified non-GMO rapeseed from Ukraine. As the used non-GMO certification is not (yet) recognised by the European Commission, the company needs to document and explain the due diligence efforts taken in this supply chain. As the company knows various European companies are buying rapeseed from the same supplier, it decides to contact the supplier´s management to find out if the supplier would be willing to undergo the process of “official recognition” of the certificate by the European Commission.
**(8.3)** If a scheme is recognised, members fulfilling the requirements of the scheme automatically comply with the EU Conflict Minerals Regulation.	The Ukrainian rapeseed supplier is able to obtain the non-GMO certification officially recognised by the European Commission. In the following years, the company does not need to invest in any additional due diligence efforts for this supply chain.
**Article 14—Guidelines**	** *Practical implications:* **
**(14.1 and 14.2)** The Commission assigns external experts to provide an indicative list of high-risk areas of GMO agriculture for each crop. The list can be used as a guide for companies carrying out due diligence.	The European Commission decides to assign external experts to compile an annual overview of agricultural goods imported (in relevant amounts) into the EU. The list contains not only the exporting countries most relevant for European companies, but also an overview of GM varieties available in these countries. It also indicates countries in which GMOs are deregulated and NGT varieties are no longer accounted as GMOs. Looking at the practical implications, a company that already has a good overview of its supply chain and has already incorporated sustainability aspects into its strategy would not require too much additional effort to comply with the proposed alternative risk-based strategy for GMO traceability.

## Data Availability

The data presented in this study are available on request from the corresponding author. The interview data are not publicly available due to privacy restrictions of the interviewees.

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
