# Peer review of "Strategies for Traceability to Prevent Unauthorised GMOs (Including NGTs) in the EU: State of the Art and Possible Alternative Approaches"

_foods, 2024, doi:10.3390/foods13030369_

Round 1

Reviewer 1 Report

Comments and Suggestions for Authors

The manuscript offers a comprehensive and well-structured examination of the complexities and prospective resolutions in the traceability of products derived from new genomic techniques (NGTs), with a particular emphasis on their integration within existing regulatory frameworks. Nevertheless, the manuscript could be further refined by addressing the following points:

1. A more detailed analysis regarding the logistical and financial implications of the proposed traceability methods would significantly enrich the paper. This is crucial, as the feasibility of these methods for companies aiming to validate their non-GMO products holds considerable commercial value. However, it is equally important to align this feasibility with the operational and cost constraints, especially under the stringent regulatory environment of the European Union.

2. The primary thrust of the article, which lies in the traceability of NGT products, does not receive the prominence it deserves in the current title. A rephrased title that better encapsulates this central theme would more effectively communicate the paper's key contribution to the ongoing dialogue surrounding NGT regulation and traceability.

3. The manuscript is marred by minor textual inaccuracies, particularly six incorrect references cite, which detract from its overall clarity and coherence. Rectifying these errors would markedly improve the manuscript's readability.

In summary, while the paper presents insightful perspectives on NGT traceability and regulatory strategies, addressing these areas of improvement would considerably augment its clarity, precision, and relevance to the field.

Reviewer 2 Report

Comments and Suggestions for Authors

Dear Authors,

The manuscript entitled “Strategies for traceability to prevent unauthorised GMO in the EU: State of the art and possible alternative approaches” deals with an interesting and current topic. However, it has some flaws that have to be handled.

The most important issue is related to the methodology and the results: it is not clear what results originate from the interviews, from the workshops, or from literature review by the authors; based on what Table 3 was compiled. Besides, more details should be provided on the interviews (when, where they were conducted) and especially on the expert workshops (where they were conducted, who the participants were, how many participants took part, etc.). It is not clear how “Voluntary Sustainability Standards” can be a stakeholder type (see Table 1), and why “Bok & Claim” method is not relevant for RTRS (see in the other Table 1).

There are some formatting issues as well, e.g., all sections are numbered 1, page numbers are not consecutive, referencing to sections, tables, and figures are erroneous (“section 0” and “Error” Reference source not found.” are mentioned several times), there are two tables numbered 1, “Tables may have a footer” is unnecessary. One citation is not formatted properly (lines 235-236).

Comments on the Quality of English Language

There are some typos and minor grammar mistakes in the text (see, e.g., lines 98, 171, 179, 187, 191, 192, 232, 233, and 413).

Round 2

Reviewer 1 Report

Comments and Suggestions for Authors

all comments were well reponsed this time, it could be suitable to pulication with final decision by editor